# The *Acinetobacter baumannii* disinfectant resistance protein, AmvA, is a spermidine and spermine efflux pump

Francesca L. Short [1,2], Qi Liu[1], Bhumika Shah [1], Heather E. Clift[1,4], Varsha Naidu [1], Liping Li [1], Farzana T. Prity[1], Bridget C. Mabbutt[1], Karl A. Hassan [3] & Ian T. Paulsen [1✉]

Antimicrobial resistance genes, including multidrug efflux pumps, evolved long before the ubiquitous use of antimicrobials in medicine and infection control. Multidrug efflux pumps often transport metabolites, signals and host-derived molecules in addition to antibiotics or biocides. Understanding their ancestral physiological roles could inform the development of strategies to subvert their activity. In this study, we investigated the response of *Acinetobacter baumannii* to polyamines, a widespread, abundant class of amino acid-derived metabolites, which led us to identify long-chain polyamines as natural substrates of the disinfectant efflux pump AmvA. Loss of *amvA* dramatically reduced tolerance to long-chain polyamines, and these molecules induce expression of *amvA* through binding to its cognate regulator AmvR. A second clinically-important efflux pump, AdeABC, also contributed to polyamine tolerance. Our results suggest that the disinfectant resistance capability that allows *A. baumannii* to survive in hospitals may have evolutionary origins in the transport of polyamine metabolites.

[1] Department of Molecular Sciences, Macquarie University, North Ryde, NSW, Australia. [2] Department of Microbiology, Biomedicine Discovery Institute, Monash University, Clayton, VIC, Australia. [3] School of Environmental and Life Sciences, University of Newcastle, Callaghan, NSW, Australia. [4] Present address: Versiti Blood Research Institute, Milwaukee, WI, USA. ✉email: ian.paulsen@mq.edu.au

Antimicrobial resistance (AMR) is a critical public health challenge of the twenty-first century. Infections caused by antibiotic-resistant bacteria are predicted to cause 10 million annual deaths by 2050 if urgent action is not taken[1]. Though extensive AMR is a clinical phenomenon, the genes responsible have deep evolutionary origins long predating the use of antibiotics in medicine[2,3]. Naturally occurring antibiotics are typically found at subinhibitory levels, while synthetic drugs and biocides are not present in natural (unpolluted) microbial environments at all. As such, AMR genes are proposed to play additional roles, for example, in bacterial signalling, metabolism and virulence[2,4,5]. Multidrug efflux pumps (MDEPs) are an important category of AMR determinant, and often have core physiological functions[6,7]. These functions broadly fall into removal of harmful exogenous molecules (for example, mammalian antimicrobial peptides or plant flavonoids) or secretion of endogenous molecules (such as siderophores, quorum-sensing signals or metabolites)[2,6,7]. Understanding the origins and ancestral functions of AMR genes, including MDEPs, could help to predict their future evolution or point to new ways to subvert their activity.

*Acinetobacter baumannii* is a notorious opportunistic pathogen classified within the 'ESKAPE' group of bacterial species that are responsible for the majority of antibiotic-resistant infections[8,9]. Most *A. baumannii* infections are caused by two dominant multidrug-resistant lineages (designated 'international clonal lineages' ICLs 1 and 2), which have spread worldwide[10]. *A. baumannii* has a particular ability to survive for prolonged times in hospital environments[11], and this is due in part to high disinfectant tolerance conferred by its repertoire of MDEPs[11,12]. Multiple *A. baumannii* MDEPs contribute to disinfectant resistance, for example, AdeABC[13], AdeIJK[13], AmvA[14], AceI[15] and AbeS[16], all confer resistance to one or both of the widely used disinfectants chlorhexidine and benzalkonium chloride. Though disinfectant resistance is an adaptive advantage for the modern clinical era, many *A. baumannii* efflux pump genes are conserved across the species or genus, suggesting a shared primordial function[2,12]. Recently it was shown that the physiological substrates of AceI—a chlorhexidine efflux pump encoded in the core genome of *A. baumannii*[15,17]—are likely to be short-chain polyamines[18].

Polyamines are an ancient class of metabolites comprising two or more amine moieties connected by aliphatic chain(s); the most common biological polyamines being putrescine, cadaverine, spermidine and spermine (Fig. 1a)[19]. These molecules have central roles in all three kingdoms of life and can be present intracellularly at high (mM) concentrations[19,20]. In bacteria, polyamines have been implicated in species-specific functions that include biofilm formation, cell growth, oxidative stress resistance and nitrogen storage, among others[21]. Many pathogenic bacteria depend on polyamine synthesis or import for their pathogenesis[22–25]. Polyamines can exert their functions in bacteria directly by virtue of their general biochemical properties or they can serve as signals, which act through specific receptors even at low concentrations[24,26]. Whether produced endogenously or present in the environment, polyamines can be toxic in high amounts. Several efflux systems have been reported to facilitate polyamine transport, such as members of the small multidrug resistance[27], major facilitator superfamily (MFS)[28], and PACE families[18].

In this study, we have investigated the transcriptomic response of *A. baumannii* to high levels of the four major biological polyamines (putrescine, cadaverine, spermidine and spermine), with a view to defining their physiological roles and identifying transporters responsible for efflux of these molecules. The efflux pump genes *aceI*, *adeABC* and *amvA* were all strongly induced by different polyamines. We show that AdeABC and AmvA are required for tolerance to the long-chain polyamines spermine and spermidine and demonstrate increased spermidine accumulation in cells lacking the MFS transporter AmvA. Finally, we also show that spermine and spermidine induce *amvA* expression and bind to its repressor, AmvR. Our results strongly suggest that long-chain polyamines are physiological substrates of the conserved *A. baumannii* efflux pump AmvA.

## Results

### RNA sequencing (RNA-Seq) of *A. baumannii* following polyamine shock

We performed transcriptomics on the widely used, ICL1 *A. baumannii* strain AB5075-UW following exposure to high levels of exogenous polyamines. The minimum inhibitory concentrations (MICs) of all molecules were very high at 10 mg ml$^{-1}$ for spermine and 40 mg ml$^{-1}$ for spermidine, putrescine and cadaverine. RNA was extracted from duplicate log-phase *A. baumannii* AB5075-UW cultures supplemented with putrescine di-hydrochloride, cadaverine di-hydrochloride, spermidine tri-hydrochloride or spermine tetra-hydrochloride at 1/8 MIC. RNA-Seq reads were mapped, normalised and fold-changes calculated as described (see 'Methods'). Each biological replicate gave rise to >10 million reads with >98% mapping to the *A. baumannii* AB5075-UW genome (Supplementary Table 1). Genes showing significantly altered expression in the presence of polyamines were defined as those with log$_2$ fold change >1 or < −1 and a corrected $p$ value <0.05 relative to the control.

### Individual polyamines induce distinct transcriptional responses

A total of 499 genes showed altered expression following treatment with one or more polyamines, with individual regulons ranging from 93 (spermidine) to 259 (spermine) genes (Fig. 1b, Supplementary Fig 1A and Supplementary Data 1). The diamines putrescine and cadaverine each caused upregulation of a large number of genes (157 and 110, respectively) and downregulation of relatively few (25 and 23). For the tetraamine spermine and triamine spermidine, the number of genes showing increased and decreased expression was more balanced (spermine: 118 up/141 down; spermidine: 55 up/38 down). Though the majority of gene expression changes were specific to just one polyamine (372 genes), there was substantial overlap in the genes regulated by putrescine and cadaverine, which had 73 common targets (Fig. 1b and Supplementary Fig 1A). Interestingly, putrescine and spermine showed divergent regulation of 21 genes. These genes included ABUW_0068–71 (involved in amino acid metabolism), ABUW_2096–2099 (fatty acid metabolism) and ABUW_2448–2456 (fatty acid metabolism). Nine genes were differentially expressed with all four polyamines: the *adeABC* efflux pump genes, the transcriptional regulator *amvR* (ABUW_1678; also called *smvR*), and ABUW_0233 were induced, the periplasmic OB-fold protein-encoding ABUW_1352 was repressed, and three genes (ABUW_2448, ABUW_2449 and ABUW-2453) were induced by putrescine, cadaverine and spermidine but repressed by spermine.

### Functional categories of polyamine-responsive genes

A summary of the clusters of orthologous groups (COG) functional classifications of polyamine-responsive genes is shown in Fig. 2a. Enrichment of specific gene ontology (GO) terms within the polyamine-regulated genes was tested using the TopGO package[29]. A summary of polyamine-regulated genes of interest is given in Supplementary Table 2.

Frequent targets included metabolic genes, particularly those involved in energy production and conversion, lipid transport and metabolism, inorganic ion transport and amino acid metabolism (COGs C, I, P and E, respectively). Transcription genes (COG K) were also common polyamine-responsive targets,

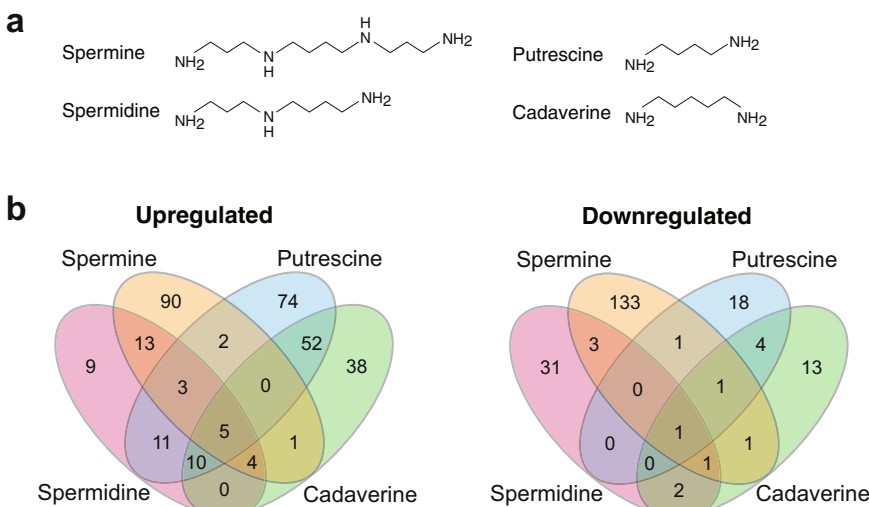

**Fig. 1 Distinct regulons of spermine, spermidine, putrescine and cadaverine. a** Structures of the four polyamines used in this study. **b** Venn diagram of upregulated and downregulated gene sets. Note that some targets showed expression changes in opposite directions following treatment with different polyamines, as shown in Supplementary Fig. 1.

particularly with spermine. As expected, given that polyamines are amidated molecules and their metabolism is tightly linked to that of amino acids, several operons involved in utilisation of amino acids were regulated by polyamine shock. This included genes for histidine utilisation (induced by putrescine and spermidine), leucine metabolism genes ABUW_2453–5 (induced by putrescine and cadaverine, repressed by spermine) and many amino acid transporters. Polyamine shock also regulated genes involved in fatty acid utilisation such as genes of the *ato* butyrate/acetoacetate degradation operon (induced by putrescine and cadaverine), ABUW_1572-4 (induced by spermidine, putrescine and cadaverine) and the ABUW_2447–58 operon for leucine and fatty acid degradation.

**Regulation of virulence genes and phenotypes in response to polyamine shock**. Virulence of *A. baumannii* depends on multiple factors, including secretion systems, pili, siderophores, efflux pumps and other defence systems[30]. Multiple genes from virulence-related pathways were induced or repressed by exogenous polyamines (Supplementary Table 2). Many iron-acquisition genes were polyamine-responsive, such as the siderophore synthesis and uptake genes induced by spermidine and spermine. In addition, polyamines downregulated the expression of several virulence-related cell surface structures including the *csu* pilus (suppressed by cadaverine), genes of the Type VI secretion system used for interbacterial competition (repressed by cadaverine and spermidine) and Type IV secretion system and competence genes (suppressed by putrescine). Polyamine-regulated stress resistance genes included those for copper resistance (induced by spermine, putrescine and cadaverine), heavy metal resistance (induced by cadaverine and spermine) and hydrogen peroxide resistance. Finally, polyamines appeared to affect the expression of genes involved in horizontal gene transfer, as well as horizontally acquired elements themselves. In addition to the Type IV pili mentioned above, which are required for horizontal gene transfer in *A. baumannii*[30], some enzymes of the CRISPR-Cas phage resistance locus were repressed by spermine, and the expression of genes within the predicted prophage regions[31] was variably affected by spermidine and putrescine (Supplementary Table 2).

To determine whether the distinct transcriptional effects of exogenous polyamines were associated with changes in virulence-related phenotypes, serum resistance and biofilm formation assays were performed (Supplementary Fig 1B, C). *A. baumannii* AB5075-UW showed a delayed susceptibility to serum killing, with a loss of viability between 1 and 3 h (Supplementary Fig 1B). Spermidine and spermine at 1/8 MIC both caused a drastic reduction in serum survival that was apparent after 30 min, while putrescine or cadaverine supplementation increased serum survival to close to 100%. *A. baumannii* AB5075-UW showed relatively low biofilm formation after 40-h static incubation in rich medium (Supplementary Fig 1B). The presence of putrescine, cadaverine or spermidine at 1/8 MIC did not affect static biofilm formation in rich media, while spermine caused a small decrease. These results do not support a major role for polyamines in biofilm regulation in *A. baumannii* AB5075-UW despite transcriptional regulation of some relevant genes (e.g. the *csu* pilus). The opposing effects of short- and long-chain polyamines on serum resistance are consistent with their divergent transcriptional regulation, though we did not notice any specific regulatory targets that would explain their effects on serum resistance.

**Multidrug efflux pumps (MDEPs) regulated by polyamine shock**. *A. baumannii* encodes a wide repertoire of MDEPs, which confer the ability to withstand many disinfectants and antibiotics. Drug efflux pumps are often induced by their substrates[6]. The efflux pump genes *aceI*, *adeABC* and *amvA* all showed increased expression in the presence of polyamines, with different combinations inducing each one (Fig. 2b). The PACE family transporter gene *aceI* was the most dramatically upregulated, with a log2-fold change of ~8 in the presence of putrescine and cadaverine and ~5 with spermine, while spermidine showed a weak (log2-fold change of 2) induction that was not statistically significant. This fits with the recent finding that short-chain diamines (including putrescine and cadaverine) are physiological substrates of AceI, while the longer-chain polyamines spermine and spermidine are not substrates but do weakly induce *aceI* expression[18]. The *adeABC* genes, encoding an RND family transporter, showed a 2–3 log2-fold increase in expression in the presence of any of the four molecules. This apparent induction by polyamines was surprising given that *A. baumannii* strain AB5075 carries a mutation in the regulator AdeS, which causes constitutive *adeABC* expression[32,33]. The MFS transporter gene *amvA* was induced by spermidine and spermine but not by putrescine and

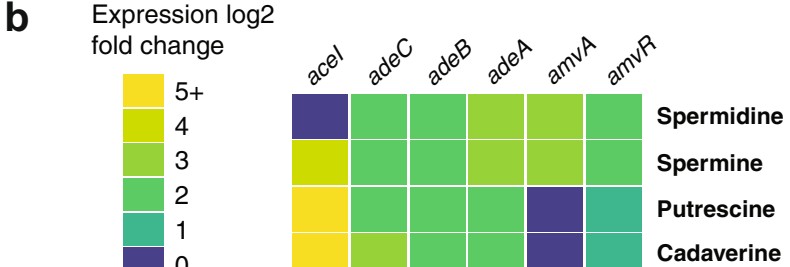

**Fig. 2 Polyamine-regulated gene functional categories and phenotypes. a** Functional categories of genes regulated by spermine, spermidine, putrescine and cadaverine. COG categories of the genes showing expression changes in response to each molecule are shown. Regulated genes came from multiple functional classes, and lipid/inorganic ion transport and metabolism genes were highly represented. **b** Discontinuous heatmap showing induction of the efflux pump genes *aceI* (PACE family), *adeABC* (RND family) and *amvA* (MFS family) and its regulator *amvR* by different polyamines.

cadaverine. The *amvA* repressor gene *amvR*[34], which is encoded 125-bp upstream of *amvA* on the opposite strand, was also induced by polyamine shock; however, the cognate regulators of *aceI* and *adeABC* (*aceR* and *adeRS*, respectively, both transcriptional activators) were not.

**AmvA is a spermine and spermidine efflux system.** As there are no characterised spermine or spermidine efflux pumps in *A. baumannii*, we sought to determine whether AmvA and/or

AdeABC may perform these roles. First, growth in the presence of polyamines was measured to test whether AmvA or AdeABC may be responsible for removing harmful levels of these molecules from the cell (Fig. 3a and Table 1). Mutation of *amvA* dramatically decreased *A. baumannii* AB5075-UW resistance to spermine (16-fold reduction in MIC) and spermidine (64-fold reduction). Mutation of *amvR*, which was previously shown to increase *amvA* expression by ~6-fold[34], increased the MIC of spermine 2-fold from 10 to 20 mg ml$^{-1}$ but had no effect on growth in the presence of spermidine. Note that the *A. baumannii*

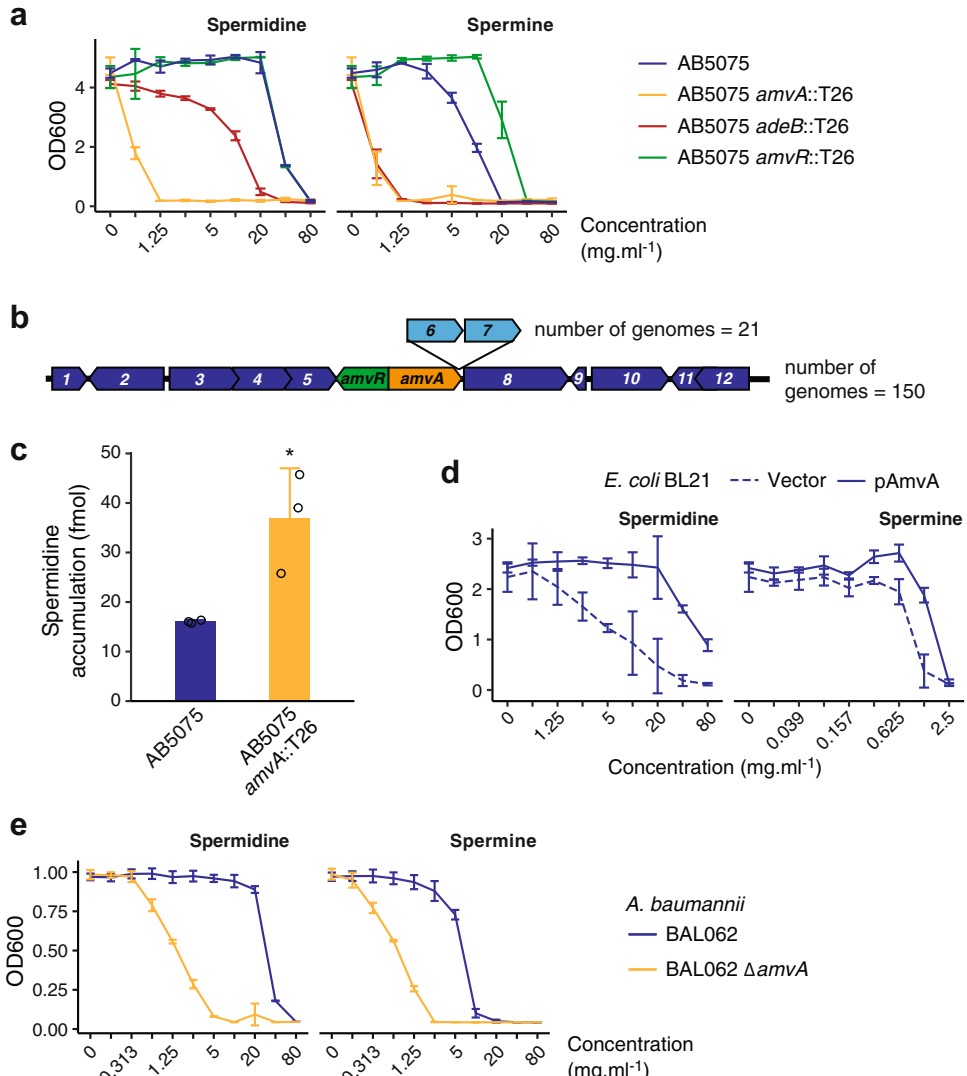

**Fig. 3 Induction of efflux pumps by polyamine shock identifies spermine and spermidine as AmvA substrates. a** Growth of *A. baumannii* AB5075 and mutants in the presence of spermine and spermidine. Results are mean ± SD for two biological and two technical replicates. **b** Genomic context of *amvAR* in 172 *A. baumannii* strains. The genes are present in 100% of complete *A. baumannii* genomes in a region of very high synteny. Core genes are shown in dark blue, accessory genes in light blue, and the number of genomes possessing each gene path is indicated. Annotated gene functions are: 1—*aceI* PACE family transporter, 2—acetyl-CoA acyltransferase, 3—short-chain dehydrogenase/reductase, 4—MaoC domain protein dehydratase, 5—beta-lactamase, 6—hypothetical protein, 7—hypothetical protein, 8—*dnaX* DNA pol III subunit tau, 9—hypothetical protein, 10—phospholipase C, 11—thioesterase family protein, 12—iron-containing alcohol dehydrogenase. **c** Intracellular accumulation of $^3$H-spermidine in *A. baumannii* AB5075 and its AB5075::*amvA* mutant. Late exponential-phase cells were washed into assay buffer and incubated with 10.8 nM $^3$H-spermidine (see 'Methods'), and the amount of accumulated radioactivity measured by liquid scintillation counting. Results shown are from three independent biological replicates. *$p < 0.05$, one-way ANOVA. **d** Growth of *E. coli* BL21 in the presence of spermidine and spermine, with or without AmvA overexpression. Results are mean ± SD for three biological and two technical replicates. **e** Growth of *A. baumannii* BAL062 and its Δ*amvA* mutant in the presence of spermidine and spermine. Results are mean ± SD for two biological and two technical replicates.

AB5075 spermidine MIC was extremely high (40 mg ml$^{-1}$)—it is possible that the *amvR* mutant does have increased AmvA-mediated spermidine efflux, but that this is not sufficient to increase the MIC >40 mg ml$^{-1}$. Alternatively, *amvA* expression may be fully derepressed with 40 mg ml$^{-1}$ spermidine such that mutation of *amvR* does not further increase expression. An *adeB* mutant showed an eightfold reduction in spermine MIC and a twofold reduction in spermidine MIC. The MICs of putrescine and cadaverine were not affected by *amvR*, *amvA* or *adeB* mutation, and none of these mutations resulted in a growth defect (Supplementary Fig 2). These observations support the hypothesis that spermine and spermidine may be the natural substrates of the well-characterised drug efflux pumps AmvA and AdeABC.

A pangenome analysis of complete *A. baumannii* genomes was conducted to explore the distribution of these efflux pumps across the species (see 'Methods', Supplementary Data 2). While *adeABC* is variably present (Supplementary Data 2 and ref. [35]), *amvA* is found in all *A. baumannii* genomes, within a region of very high synteny comprised of other conserved genes (Fig. 3b). Due to its very strong phenotype, and its absolute conservation hinting at a deeply rooted role in *A. baumannii* biology, AmvA was selected for further characterisation.

An *amvA* mutant of *A. baumannii* AB5075 was tested against a selection of its previously reported substrates[14] in order to compare its effect on resistance to these molecules with its effect on polyamine resistance (Table 1). Mutation of *amvA* did not

**Table 1 Polyamine MIC changes in *amvA*, *amvR* and *adeB* mutants of *A. baumannii* AB5075.**

| | MIC (mg ml⁻¹) [fold change] | | | |
|---|---|---|---|---|
| Polyamines | WT | *amvA*::T26 | *amvR*::T26 | *adeB*::T26 |
| Spermine | 10 | 0.625 [16] | 20 [0.5] | 1.25 [8] |
| Spermidine | 40 | 0.625 [64] | 40 | 20 [2] |
| Putrescine | 40 | 40 | 40 | 40 |
| Cadaverine | 40 | 40 | 40 | 40 |
| | MIC (µg ml⁻¹) [fold change] | | | |
| Reported AmvA substrates | WT | *amvA*::T26 | | |
| Acriflavine | 64 | 16 [4] | | |
| Benzalkonium chloride | 32 | 32 | | |
| Chlorhexidine | 15 | 7.5 [2] | | |
| Erythromycin | 250 | 250 | | |
| Ethidium bromide | 32 | 32 | | |
| Methyl viologen | 100 | 25 [4] | | |

affect the MIC of three of the six substrates tested, while three substrates—acriflavine, chlorhexidine and methyl viologen—showed a modest (twofold or fourfold) MIC reduction in AB5075 *amvA*::T26. Note that previous investigation of the AmvA substrate range used a drug-sensitive *A. baumannii* strain AC0037; AB5075-UW has a different complement of efflux pumps that could potentially mask some AmvA activities. Overall, AmvA has a much greater impact on resistance to toxic levels of polyamines than on resistance to its previously reported substrates.

We then tested whether loss of *amvA* affects accumulation of spermidine in cells (Fig. 3c). *A. baumannii* AB5075-UW and *A. baumannii* AB5075-UW *tn-amvA* were grown to exponential phase, washed, and incubated with 10.8 pmol [³H]-spermidine. Mutation of *amvA* significantly increased the amount of intracellular [³H]-spermidine, though accumulation of exogenous spermidine was very low in the conditions tested (approximately 15 fmol in $2 \times 10^8$ cells), perhaps because of the low concentration of radioactive substrate used, because the cells retained a functional AdeABC efflux system, or because the accumulation of spermidine is likely to rely largely on the activity of uptake systems.

Following the finding that AmvA influences both polyamines tolerance and accumulation in *A. baumannii* AB5075, we wished to determine whether AmvA could promote spermidine and spermine resistance in a bacterium lacking this MDEP. *Escherichia coli* BL21 cells overexpressing AmvA showed full growth in the presence of spermidine up to 20 mg ml⁻¹, while growth of the vector-only control strain decreased at ≥2.5 mg ml⁻¹ spermidine (Fig. 3d). AmvA also increased resistance to spermine to a lesser extent (Fig. 3d).

Finally, we examined the effect of *amvA* mutation on a distinct *A. baumannii* strain, BAL062, which is a representative of the ICL2 lineage (AB5075-UW is an ICL1 strain). Similarly to our findings in AB5075-UW, the BAL062 Δ*amvA* mutant showed dramatically reduced tolerance to exogenous spermidine (32-fold MIC reduction) and spermine (8-fold MIC reduction) (Fig. 3e).

**AmvR is a spermine and spermidine-responsive repressor of *amvA*.** We next aimed to find out whether expression of *amvA* is regulated by its polyamine substrates and whether AmvR is responsible for this regulation. Expression of *amvA* was measured following exposure to spermidine or spermine at 1/16 MIC (~10 and ~2 mM, respectively) for 30 min, in both wild-type and *amvR* mutant *A. baumannii* AB5075 (Fig. 4a). In the wild-type strain, *amvA* expression increased ~6-fold with spermidine and ~2-fold

with spermine. The *amvR* mutant strain showed a ~10-fold increase in the expression of *amvA* relative to the wild-type strain, and the expression did not increase further on polyamine treatment. Regulation of *amvA* in AB5075 was also explored using a reporter fusion vector comprising the *amvR-amvA* intergenic region upstream of green fluorescent protein (GFP; Supplementary Fig 3). Expression of GFP was significantly higher in AB5075 *amvR*::T26 than the wild-type strain (Supplementary Fig 3A) and did not increase further with spermidine treatment (Supplementary Fig 3B). The *amvA*prom-GFP reporter was strongly induced by 10 mM spermidine but not by putrescine or cadaverine at 10 mM (Supplementary Fig 3C), while spermine was toxic at this concentration. Note that the polyamine *amvA*prom-GFP induction experiments were conducted in *A. baumannii* ATCC 17978 (a widely used antibiotic-sensitive strain) using a gentamicin-resistant vector backbone, due to slight toxicity and high baseline activity of the zeocin-resistant vector used in AB5075 (Supplementary Fig 3A, B). Taken together, these results demonstrate that AmvR is a repressor of *amvA*, but AmvR repression is lifted in the presence of long-chain polyamines.

We then tested whether AmvR has binding capacity for long-chain polyamines, with the aim of finding out if long-chain polyamines may induce *amvA* expression by direct binding of its repressor. Purified AmvR carrying a C-terminal hexahistidine tag was characterised by analytical size-exclusion chromatography (SEC) to be ~50 kDa in solution, consistent with its native, stable dimeric form (Supplementary Fig 4A). AmvR was first screened for interaction with polyamines using differential scanning fluorimetry (DSF). This method monitors changes to the thermal stability of a protein, observed as a dye response during unfolding, to indicate potential ligand binding[36]. Spermine and spermidine at 0.2% (5.7 and 7.8 mM, respectively) both elevated the thermal stability of AmvR by +4 °C while putrescine caused little change (+1 °C shift, i.e. below the significance threshold of the DSF experiment).

We then utilised a nano DSF (nDSF)-based isothermal approach to determine the binding constant ($K_d$) of AmvR for spermine and spermidine. The label-free nDSF assay monitors intrinsic tryptophan and tyrosine fluorescence upon protein unfolding, and using unfolding curves at varying ligand concentrations, the isothermal analysis of nDSF data facilitates $K_d$ determination by fitting the unfolded protein fraction vs. ligand concentrations at a selected temperature[37,38]. As shown in Fig. 4b, the isothermal analysis of the nDSF data for AmvR/spermine and AmvR/spermidine at 49 °C (heating rate of 2 °C min⁻¹) yields a low micromolar affinity, with $K_d$ of 5.4 and 9.5 µM, respectively (the change of heat capacity, $\Delta C_p$ fitted). In contrast, isothermal analysis of the nDSF data for AmvR/putrescine at 51 °C yields $K_d$ of 260 µM. Using the $\Delta C_p$ set to zero instead of fitted $\Delta C_p$ did not result in any differences in the measured $K_d$ in this experiment (Supplementary Tables 3 and 4). Binding affinities were also measured in a replicate experiment with a slightly faster heating rate (Supplementary Fig 4C and Supplementary Tables 3 and 4) and showed minimal differences.

These specific responses indicate that, in isolation, AmvR can form complexes with spermine or spermidine as ligands with low micromolar affinity, supporting the model of AmvR as a polyamine-responsive repressor of *amvA*.

## Discussion

Bacterial drug efflux pumps can have important physiological roles in addition to their resistance functions[6,39]. Here we have investigated the effects of four of the predominant biological polyamines on *A. baumannii* and identified a role for one of its major efflux pumps, AmvA, in the transport of long-chain

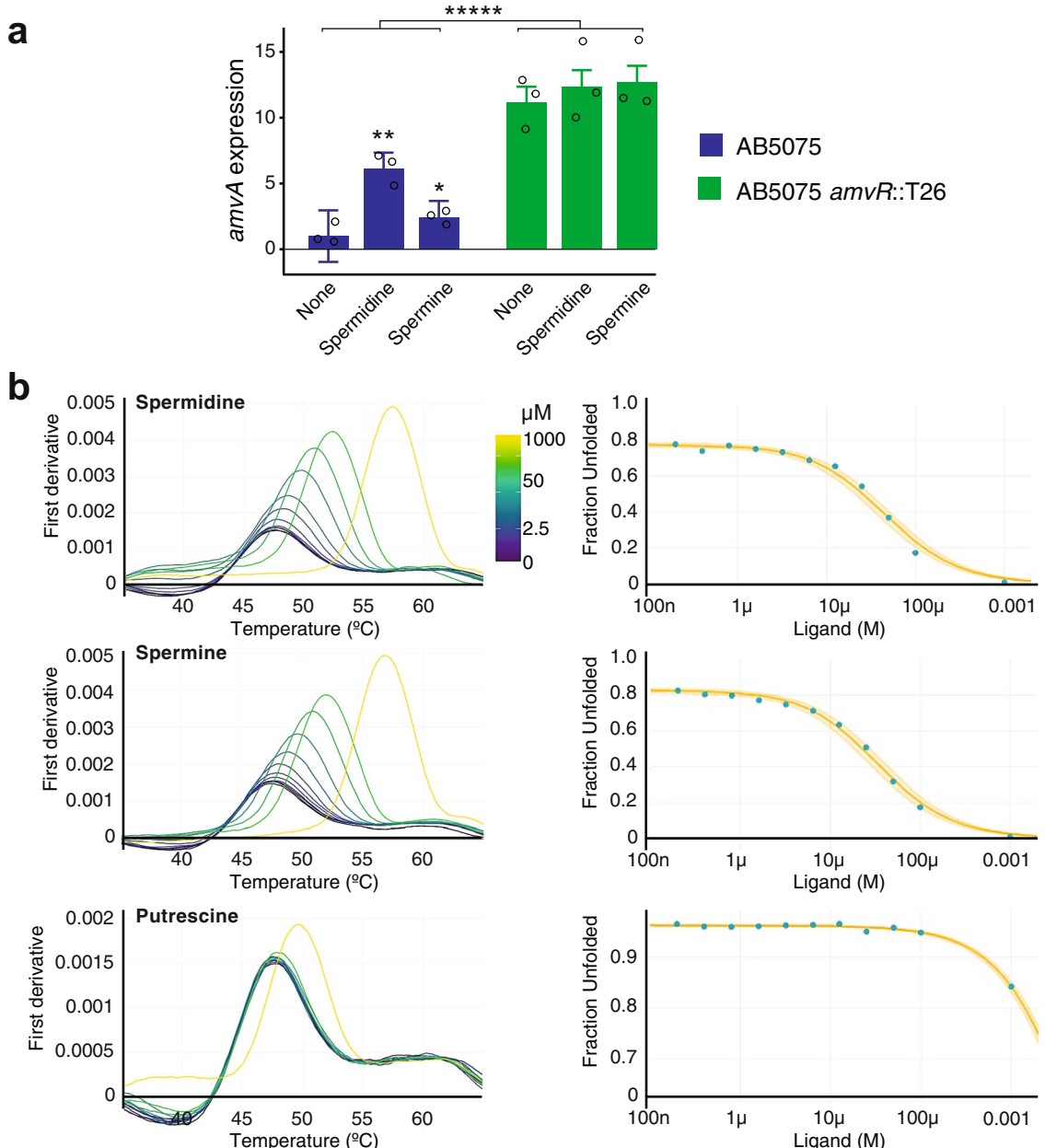

**Fig. 4 AmvR and long-chain polyamines control *amvA* expression. a** AmvR-dependent induction of *amvA* by long-chain polyamines. Quantitative real-time PCR of *amvA* transcript in AB5075 and AB5075 *amvR*::T26 following addition of long-chain polyamines at 1/16 WT MIC (10 mM spermidine, 2 mM spermine). AB5075 *amvR*::T26 showed dramatically increased *amvA* transcript levels, which did not increase further on polyamine addition. All fold changes are expressed relative to the untreated AB5075 control. Results shown are the geometric mean ± standard deviation of three biological replicates, which were each comprised of two technical replicates. A two-factor ANOVA showed a significant effect of strain background (*****$p < 0.00001$) and polyamine treatment ($p = 0.0013$, comparison not shown) on *amvA* expression levels. A one-way ANOVA with Dunnett's post hoc test was used to compare polyamine-treated samples with the untreated sample of the same strain (*$p < 0.05$, **$p < 0.01$). **b** Nano-DSF analysis of purified AmvR shows spermine and spermidine binding at low µM affinity. Melt curves and fitted unfolded protein fraction are shown for AmvR in the presence of increasing polyamine concentrations. Spermidine and spermine prevented protein unfolding at low concentrations (<10 µM), while putrescine only influenced protein unfolding at the highest concentration of 1 mM. The complete experiment was performed in duplicate, and results from the second replicate are provided in Supplementary Fig. 4. Calculated binding affinities are provided in Supplementary Tables 3 and 4.

polyamines. To our knowledge, this is the first spermidine and spermine efflux system to be identified in *A. baumannii*.

Our findings support the hypothesis that polyamines have broad biological roles in *A. baumannii*. Four hundred and ninety-nine genes had altered expression (representing 1/6 of the annotated CDS in AB5075), including many virulence-associated loci such as those encoding siderophores, pili and secretion systems (Figs. 1 and 2, Supplementary Fig. 1A, Supplementary

Table 2 and Supplementary Data 1). A limitation is that our transcriptomics was performed on cells shocked with high levels of exogenous polyamines, ranging from 18 mM (spermine) to 56 mM (putrescine). These concentrations were selected to show the full range of polyamine-responsive genes and to elicit strong induction of candidate transporters but are higher than reported intracellular concentrations of polyamines in Gram-negative bacteria (e.g. spermine ~6 mM, putrescine ~20 mM[40]) or in

most host environments. The transcriptional responses defined here could arise from metabolic feed, biophysical modulation of DNA or RNA structure, activation of specific regulatory pathways or a combination of all three. Experiments at physiological polyamine levels would help to define the biological roles of these molecules in more detail and determine whether *A. baumannii* utilises polyamines as specific virulence-regulating signals, as shown in some other bacterial species[24,26].

Several lines of evidence indicate that long-chain polyamines are biological substrates of AmvA: (1) loss of this efflux pump dramatically reduced spermine and spermidine tolerance in both ICL-1 and ICL-2 *A. baumannii* strains, (2) loss of *amvA* increased intracellular spermidine accumulation, (3) *amvA* overexpression in *E. coli* increased long-chain polyamine resistance, (4) *amvA* expression is increased in response to these molecules, and (4) regulation of *amvA* depends on its cognate regulator, AmvR, which binds to long-chain polyamines with low µM affinity. Note that *amvA* expression is not induced by its other substrates, including ciprofloxacin[41], deoxycholate[42], chlorhexidine[15] or benzalkonium chloride[43].

AmvA is clinically important in *A. baumannii*; it confers resistance to widely used disinfectants (e.g. chlorhexidine, benzalkonium), and in hospital-adapted strains, its expression is increased[14]. AmvA is a member of the DHA-2 (drug H$^+$ antiporter-2) subfamily of MFS transporters and the first member of this family shown to transport long-chain polyamines. DHA-2 proteins comprise 14 transmembrane helices[44] and have a large, central hydrophobic cavity with several acidic residues that facilitate both proton and substrate binding[7,45,46] and determine substrate charge specificity[47]. The substrate profile of AmvA includes a range of hydrophobic, cationic compounds with different degrees of protonation at physiological pH[14,48]. Spermine and spermidine are both long, flexible hydrophobic molecules with multiple protonated sites at physiological pH—it is likely that their transport requires particular active site properties, seen in DHA-2 proteins, that would also permit efflux of other substrates. Note that spermidine efflux has been demonstrated for the more distantly related DHA-1 transporter Blt, of *Bacillus subtilis*[28], which was also originally characterised for its drug-resistance activity[49].

We propose that the clinically relevant anti-biocide activity of the well-known *A. baumannii* efflux pump AmvA stems from an ancestral role in homoeostasis of long-chain polyamines. Interestingly, multiple bacterial pathogens including *Staphylococcus aureus*, *Proteus mirabilis* and members of the *Enterobacteriaceae* possess AmvA homologues (QacA, SmvA), which, like AmvA, confer increased biocide resistance (particularly to chlorhexidine) and are upregulated or have increased prevalence in clinical strains[50–53]. AmvA shares 50–55% amino acid identity with the enterobacterial SmvA proteins, while *S. aureus* QacA is more distantly related (30% amino acid identity). It is tempting to speculate that some of these clinically important AmvA homologues may also have physiological roles in polyamine transport.

A further question is why *A. baumannii* would require specific efflux systems for spermine and spermidine? Such systems may protect from exogenous polyamines encountered in host environments. Polyamine levels can reach the mM range in plants, the gut, and in serum of patients suffering from some diseases[54–57]. In particular, a recent study showed serum levels of spermidine and spermine at >500 µg ml$^{-1}$—close to their respective MICs in AB5075-UW *amvA* or *adeB* mutants (Fig. 3b)[57]. Alternatively, efflux of endogenous polyamines by AmvA (and perhaps AdeABC) may provide adaptive advantages unrelated to detoxification. For example, rapid putrescine efflux helps to counter osmotic stress in *E. coli*[58]. *A. baumannii* produces a range of

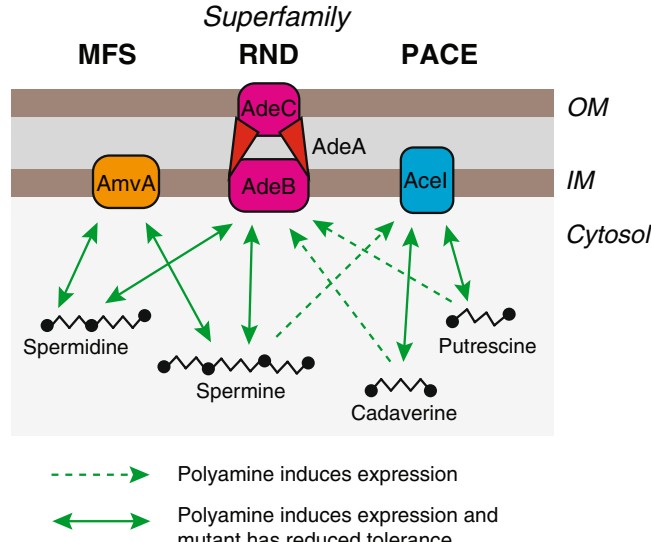

**Fig. 5 Schematic of known links between polyamines and efflux pumps in *A. baumannii*.** AmvA, AdeABC and AceI of the MFS, RND and PACE efflux pump families are each induced by, and confer tolerance to, distinct subsets of the four major biological polyamines.

endogenous polyamines, including spermidine[59–62]; however, their biological role(s) are not yet fully understood.

The ability of *A. baumannii* to withstand biocides and antibiotics is a major barrier to the effective control of this pathogen. Here we have shown that a key biocide resistance determinant, AmvA, is a spermidine and spermine efflux system and provide preliminary evidence that AdeABC may share this activity. Together with previous findings on AceI[18], our work shows that three well-known *A. baumannii* efflux pumps, of three distinct transporter families, function in polyamine transport. AmvA, AdeABC and AceI are each induced by, and provide tolerance to, different combinations of polyamines (Fig. 5), and we speculate that polyamine tolerance in *A. baumannii* is determined by the respective expression levels, substrate profiles and transport capacity of these three efflux systems. All three of these polyamine tolerance determinants also provide some resistance to the synthetic biocide chlorhexidine, which may be a common secondary substrate among polyamine efflux systems. Overall, our results suggest a strong link between homoeostasis of the ancient, highly abundant polyamine class of molecules and the broad and flexible efflux pump activity that allows bacterial pathogens—including *A. baumannii*—to survive treatment with disinfectants.

## Methods

**Bacterial growth**. Strains, plasmids and oligonucleotides used in this study are given in Supplementary Tables 5–7. *A. baumannii* was cultivated in Muller–Hinton (MH) broth at 37 °C for all experiments with the exception of reporter assays in *A. baumannii* AB5075-UW pFLS45, which were conducted in low-salt LB. When necessary, media were supplemented with polyamine salt compounds: spermidine trihydrochloride, putrescine di-hydrochloride, cadaverine di-hydrochloride or spermine tetra-hydrochloride. Antibiotics used were zeocin (150 µg ml$^{-1}$ for *A. baumannii*; 25 µg ml$^{-1}$ *E. coli*), gentamicin (10 µg ml$^{-1}$), kanamycin (50 µg ml$^{-1}$) and ampicillin (50 µg ml$^{-1}$).

**Antimicrobial susceptibility testing**. MICs were determined by broth dilution[63] in 96-well plates with a final volume of 200 µl (AB5075, *E. coli*) or 100 µl (BAL062). Polyamine salts were added to the MH media (buffered to pH 7.8 using HEPES salt) to final concentrations of 80 mg ml$^{-1}$. Plates were incubated at 37 °C overnight and end-point growth at OD$_{600}$ measured. *A. baumannii* strains were used in MIC experiments directly from overnight cultures, at an inoculum of ~10$^6$ cells ml$^{-1}$, and plates were incubated without shaking. *E. coli* BL21 cells carrying either pTTQ18R6SH6 or pTTQ18R6SH6-AmvA (previously called

AedF)[48] were prepared as follows: cells were grown overnight in MHII, subcultured at 1:10 and induced with 0.05 mM IPTG for 1 h at 37 °C 200 rpm, and used to inoculate MIC broth dilution plates at approx. $2.5 \times 10^5$ cfu ml$^{-1}$. Ampicillin and 0.05 mM IPTG were included in media to maintain *amvA* expression, and plates were sealed with breathable film and incubated with shaking for 24 h prior to OD$_{600}$ measurement.

**RNA-Seq and analysis**. *A. baumannii* AB5075-UW cells were grown overnight and subcultured in MHII broth and grown to mid-exponential phase (5 ml, OD$_{600}$ = 0.6). Cultures were treated with polyamine salts (1/8 MIC) for 30 min at 37 °C, with shaking. Control cultures contained no polyamine salt compounds. Cell pellets were recovered (5000 g, 15 min at 4 °C) and lysed in QIAzol reagent (Qiagen). Total RNA was isolated using the RNeasy Mini Kit (Qiagen) according to the manufacturer's instructions and DNA was removed with DNaseI (TURBO DNA-free™ Kit, Invitrogen). Removal of rRNA and RNA-Seq were conducted by the Ramaciotti Centre for Genomics (Sydney, Australia) as follows: rRNA was depleted using Ribozero (Invitrogen), libraries were prepared using a TruSeq Stranded RNA-seq Prep Kit (Illumina) and sequencing was performed on an Illumina NextSeq machine with a read length of 75 bp. Sequencing statistics are given in Supplementary Table 1.

The raw data from the samples were analysed using EDGE-PRO with Bowtie 2[64]. Reads were normalised and differential expression analysis was conducted with the DESeq R package[65]. A negative binomial model was used to test the significance of differential expression between control cells and polyamine-treated cells. Genes with adjusted *p* values <0.05 and presenting at least 2-fold differences in expression were considered to be differentially expressed.

Functional enrichment analyses were conducted by first assigning COG and GO terms to AB5075-UW genes using EggNOG mapper[66]. Significantly enriched GO terms within the biological process, molecular function and cellular compartment ontologies were identified using the TopGO package with the 'weight01' algorithm and Fisher's exact test.

**Serum resistance assays**. Bacteria were grown overnight in MHII broth, subcultured 1:100 in fresh medium and grown to late exponential phase (OD$_{600}$ = 1). Cultures were then washed once in phosphate-buffered saline (PBS) and diluted 1:100 in sterile PBS; 50 μl diluted culture was added to 100 μl pre-warmed human serum (Sigma) and incubated at 37 °C. Samples were taken at set time points, serially diluted and plated for enumeration of viable bacteria.

**Biofilm formation assays**. In all, 2.5 ml cultures of *A. baumannii* AB5075-UW with or without polyamine supplementation were grown for 40 h at 37 °C in 15 ml polystyrene tubes. Attached biofilm in the tubes was quantified by crystal violet staining as described[67].

**Mutagenesis of *A. baumannii* BAL062**. The *A. baumannii* BAL062 Δ*amvA* strain was constructed by linear DNA transformation[68]. Cells were prepared for transformation by washing an overnight culture three times in 10% (w/v) sucrose, then resuspending to a final OD$_{600}$ of 20. In all, 200 μl of cell suspension was mixed with 3 μg DNA fragment and electroporated (cuvette gap 2 mm, electroporation parameters: 2500 kV, 200 Ω, 25 μF, recovery 2 h 37 °C in 1 ml LB).

**Growth curves**. For 96-well plate-format growth curves, individual wells were inoculated with 150 μl of MHII medium seeded with the strain of interest at an OD$_{600}$ of 0.05. The plate was sealed with gas-permeable film (Aeraseal™) and incubated in a Tecan Infinite 200 plate reader at 37 °C with shaking and OD$_{600}$ measurement every 15 min. For aerobic growth curves, 5 ml of MHII medium in a 50 ml flask was inoculated at OD$_{600}$ 0.025 and cultures were grown at 37 °C with continuous shaking at 200 RPM.

**Pangenome analysis**. All complete *A. baumannii* genome assemblies were downloaded from https://www.ncbi.nlm.nih.gov/assembly/ on March 26, 2020. Complete assemblies were used to minimise the effects of contig boundaries on gene neighbourhood analysis. Genome sequences were annotated using prokka[69] version 1.14.6 with the following commands passed to the program: 'prokka --prodigaltf prodigal_training.txt --proteins AB5075_UWversion.gb --genus Acinetobacter --species baumannii'. The training model was generated using default settings, and the pipeline set to preferentially assign genes based on the *A. baumannii* AB5075-UW genome annotation. Gene content analysis across the population was conducted using Panaroo[70] version 1.1.2 with core gene alignment. Gene neighbourhood information was extracted from the final pangenome network using the get_neighbourhood script distributed with Panaroo. The gene presence–absence table, which includes accession IDs for the genomes examined, is provided as Supplementary Data 2.

**Spermidine transport assays**. *A. baumannii* AB5075-UW and AB5075-UW tn-*amvA* were grown overnight in MHII broth, then diluted 1:100 in the same medium and grown to late exponential phase (OD$_{600}$ = 1.0). Spermidine was added at 1/16 MIC to induce efflux pump expression, and cultures were incubated for a further

30 min. Cells were washed twice in assay buffer (20 mM HEPES pH 8.0, 145 mM NaCl, 5 mM KCl) and resuspended to a final OD$_{600}$ of 1.0 in assay buffer supplemented with 0.5% (w/v) succinate to energise the cells. Accumulation assays were performed at room temperature with 1 ml resuspended cells. Bacterial samples were incubated with 10.8 pmol $^3$H-spermidine (46.1 mCi mmol$^{-1}$, Perkin-Elmer) for 2 min, then 200 μl of each reaction was immediately filtered through a 0.2 μM nitrocellulose membrane and washed with 2 ml assay buffer. Filters were placed in 5 ml PE Emulsifier-Safe scintillation fluid, and the amount of radiation retained on each filter (indicating intracellular $^3$H) was measured by liquid scintillation.

**DNA manipulations**. The linear fragment for mutagenesis of *A. baumannii* BAL062 to remove its *amvA* gene was constructed by overlap PCR using primers AP2064 and FS175–179 (Supplementary Table 7) to generate a DNA fragment comprising the kanamycin resistance gene from pCR2.1 flanked by the 1000 bp sequences 5′ and 3′ of *amvA*.

To construct the reporter vector pFLS43 and pFLS45, the GFP marker from plasmid pDiGc was edited by overlap PCR using FS9 + FS10 (outer primers) and FS65 + FS66 (overlap primers) to remove its internal NdeI site and introduced into pCR2.1 by TOPO cloning (Invitrogen). The *amvA-amvR* intergenic region from *A. baumannii* AB5075-UW was amplified using primers FS77 + FS78 and cloned upstream of the GFP-terminator fragment from pDiGc using the NdeI site. The promoter-GFP insert was subcloned from pCR2.1 into pVRL1 and pVRL1-Z using restriction enzymes SacI and XhoI. The AmvR expression vector was constructed by amplifying *amvR* from *A. baumannii* ATCC17978 genomic DNA using primers AmvR-F and AmvR-R and cloning into the expression vector pTTQ18$_{RGSH6}$.

**Quantitative real-time PCR (qPCR)**. Three biological replicate overnight cultures each of AB5075 wild type and AB5075 *amvR*::T26 were prepared from single colonies on freshly streaked MH agar plates. Subcultures were grown in MHII medium to OD$_{600}$ 0.5–0.6 and treated with either 2 mM spermine, 10 mM spermidine or no additive for 30 min. The cultures were pelleted at $7000 \times g$ 4 °C for 5 min. Total RNA was extracted by the miRNeasy Mini Kit (Qiagen) according to the manufacturer's instructions and residual genomic DNA was removed by DNase I (TURBO DNA-*free* Kit, ThermoFisher). KAPA SYBR FAST One-Step Kit (Merck) was used for cDNA synthesis and qPCR in LightCycler 480 II (Roche) with the following cycling parameters: cDNA synthesis and reverse transcriptase (RT) inactivation—42 °C 5 min–95 °C 2 min; qPCR cycling—(95 °C 3 s–60 °C 20 s) × 40; dissociation—60–95 °C at 30 s °C$^{-1}$. Expression of *amvA* was measured using the ΔΔCt method with *gadph* as the reference gene. Primer sequences are provided in Supplementary Table 7.

**Promoter fusion experiments**. Reporter plasmid pFLS45, comprising GFP under the control of the *amvA* promoter, was transformed into *A. baumannii* AB5075-UW and *A. baumannii* AB075 tn-*amvR* and maintained with zeocin selection in low-salt LB. Reporter plasmid pFLS43, which has the same structure but with gentamicin resistance, was transformed into *A. baumannii* ATCC17978 and maintained in gentamicin-supplemented MHII medium. Expression without induction (Supplementary Fig 3A) was measured following growth in 10 ml cultures to an OD of 2.0. Induction by polyamines was measured by growing cells to late exponential phase in 10 ml cultures, then transferring cultures to a 96-well plate format (100 μl per well), adding polyamine salts, incubating the plate at 37 °C and measuring fluorescence and OD over time. For all experiments, expression was determined by measuring culture fluorescence (ex 485/em 520) and OD$_{600}$ in a 96-well plate in a Pherastar plate reader. GFP fluorescence/OD was calculated for each well, with cells lacking the reporter vector used as a background control. Results shown represent three biological replicates and six technical replicates.

**Expression and purification of AmvR**. AmvR protein was expressed by growing BL21 pAmvR cells in ZYP-rich autoinduction medium[60] to an OD of 1.2–1.4 at 25 °C (approx. 24 h incubation). Recovered cells were resuspended in 30 ml AmvR buffer (50 mM HEPES pH 8.0, 300 mM NaCl) supplemented with 5 mM imidazole and 5% glycerol and stored at −80 °C. Cell aliquots were thawed in the presence of lysozyme (1 μg ml$^{-1}$) and DNase I (5 μg ml$^{-1}$) and lysed by cell disruptor (Constant Systems) at 30 KPsi. Clarified cell lysates (following 40 min centrifugation at $11,000 \times g$ and 0.2 μm filtration) were loaded with a peristaltic pump onto pre-packed columns (1 ml) of Ni-Sepharose media (His trap, GE Healthcare) pre-equilibrated in the same buffer. The column was then washed with 50 column volumes of AmvR buffer + 50 mM imidazole, and protein was eluted in AmvR buffer + 500 mM imidazole. Eluate fractions were pooled and further purified using SEC (Superdex HiLoad 200 16/600 column, GE Healthcare) in AmvR buffer (50 mM HEPES pH 8.0, 300 mM NaCl) supplemented with reducing agent tris(2-carboxyethyl)phosphine (TCEP, 1 mM) and glycerol (5% v/v). Protein-containing fractions were pooled, concentrated using centrifugation (Centricon 10 kDa molecular weight (MW) cut-off) and snap-frozen in liquid N$_2$. The purity of the recovered His$_6$-tagged product was verified by using sodium dodecyl sulfate–polyacrylamide gel electrophoresis, showing a single band at ~25 kDa when visualised with Coomassie blue dye. MW of AmvR in solution was estimated using analytical SEC procedures on a Superdex 200 10/300 GL column (GE-Healthcare),

by comparing its elution volume to those of a set of MW standards (HMW and LMW calibration sets, GE Healthcare).

**Differential scanning fluorimetry**. DSF was performed according to standard methods[36]. SYPRO Orange (100×, Invitrogen) was diluted into AmvR buffer and mixed with purified AmvR (final concentration 1 mg ml$^{-1}$). Twenty-microlitre aliquots of AmvR-SYPRO orange mixture were transferred to a 96-well plate in triplicate, polyamine salts were added at 0.2% (w/v), which corresponds to 5.7 mM spermine and 7.85 mM spermidine, and the solutions were gently mixed by plate centrifugation (1000 RPM, 1 min). The plate was heated at 1 °C min$^{-1}$ over 25–95 °C in a real-time qPCR machine (Mx3005P, Stratagene) and change in fluorescence intensity (dR, automatically baseline corrected) at 610 nm was measured. Derivative curves were used to calculate the transition midpoint and assign melting temperature.

**nDSF binding study**. Ligand interactions for AmvR (15 µM) were monitored in duplicate in HEPES buffer (50 mM, with 300 mM NaCl, pH 8.0) supplemented with TCEP (1 mM) and glycerol (5% v/v). Potential ligands (spermidine, spermine, putrescine) were tested from 0.2 to 100 µM following 2-fold serial dilutions in the same buffer. Additional end-point measurements were made at 1 mM for each compound. Following room temperature incubation (15 min), samples were transferred to standard grade capillaries (Prometheus NT.48 series, Nanotemper). Samples were heated over 20–95 °C at 2 °C min$^{-1}$ (replicate 1) or 1.5 °C min$^{-1}$ (replicate 2) with a Prometheus NT.48 fluorimeter (Nanotemper) controlled by PR.ThermControl. Excitation power was pre-adjusted to obtain fluorescence readings >2000 relative fluorescence units for emission at 330 nm (F330) and 350 nm (F350).

The isothermal analysis of nDSF data was performed using the web server FoldAffinity[38] utilising the approach of Bai et al.[37]. Briefly, each fluorescence curve was fitted by six free parameters (melting temperature $T_m$, unfolding enthalpy $\Delta H$ and initial and final intercepts and slopes). Curve fitting was subsequently improved by a global fit of slopes in conjunction with other parameters still fitted individually to curves. From best fits, the fraction of unfolded protein was determined for each ligand concentration for one or several temperatures of interest. Finally, the fraction unfolding vs. ligand concentrations were incorporated into a 1:1 binding model for construction of isothermal plots. All isothermal analysis utilised change of heat capacity ($\Delta C_p$) values suggested by the FoldAffinity server[38], as well as by setting $\Delta C_p$ to zero. The temperature with the lowest numerical fitting error of binding constant ($K_d$) was selected for the isothermal plots shown in this study.

**Statistics and reproducibility**. All microbiological experiments reported in the manuscript are from three biological replicates, defined as samples originating from separate overnight cultures each started from a single, separate colony, with technical replicates included as appropriate. Technical replicates were used for qRT-PCR quantification, and for all MIC, growth curve and reporter gene expression experiments were performed in a 96-well plate format and were defined as reactions performed on the same cDNA sample (for qRT-PCR experiments) or wells seeded with the same culture (for plate-format bacteriological experiments). Statistical significance for microbiological experiments was, in most cases, determined by analysis of variance with an appropriate post hoc test. For qRT-PCR experiments, where results are expected to be log-distributed, data were log2-transformed prior to statistical analysis and geometric means are reported. Specific numbers of replicates for each experiment (biological and technical), and details of the specific statistical test(s) used, are provided in the figure legends. In experiments where both biological and technical replicates were used, the mean of the technical replicate values derived from the same biological replicate was used for subsequent analyses.

**Reporting summary**. Further information on research design is available in the Nature Research Reporting Summary linked to this article.

## Data availability
Raw RNA sequencing data associated with this study are available from the European Nucleotide Archive, accession PRJEB40527. Source data for figures is provided as Supplementary Data 3.

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

## Acknowledgements

We thank members of the Paulsen laboratory for useful discussions. We thank the Ramaciotti Centre, University of New South Wales for performing RNA sequencing. We also gratefully acknowledge the use of equipment in the Structural Biology Facility within the Mark Wainwright Analytical Centre, University of New South Wales, funded in part by the Australian Research Council Linkage Infrastructure, Equipment and Facilities Grant: ARC LIEF 190100165. This work was funded by grant 1120298 from the National Health and Medical Research Council of Australia. KAH is supported by an Australian Research Council Future Fellowship FT180100123. FLS is supported by an Australian Research Council DECRA fellowship DE200101524.

## Author contributions

K.A.H. and I.T.P. conceived and designed the study. Q.L. performed RNA sequencing experiments and initial analysis. B.S. performed nanoDSF experiments, and H.E.C. developed the AmvR purification protocol and performed initial DSF experiments, with supervision from B.A.M. V.N. performed exploratory MIC measurements. L.L. and F.T.P. performed qRT-PCR experiments. F.L.S. completed all other experiments and analyses and wrote the manuscript, with input from all other authors.

## Competing interests

The authors declare no competing interests.
