## [Transparent Peer Review File · Communications Biology]

Reviewers' comments:

Reviewer #1 (Remarks to the Author):

Short et al., describe a thorough study intending to delineate the cellular transcriptional response of *A. baumannii* to high levels of polyamines. Through this process, they newly insinuate AmvA, and adeABC, as spermidine/spermine efflux pumps (long-chain polyamines). The impact of such a finding is significant since there are only a handful of drug efflux pumps that have been associated with this phenomenon. Furthermore, the link between polyamines and disinfectant resistance is compelling in terms of evolutionary significance. Below I have provided feedback to improve the study (L=line). Overall, the transcriptomic approach is a strength, and the evidence supporting AmvA's association with polyamine export is compelling. I have concerns regarding the suggestion that polyamines bind to AmvR, and thus alleviate the protein's repression of AmvA. While this may indeed be the case, this section of the study needs attention (as described below) and likely additional investigation before this is conclusively stated. The assay involving spermidine accumulation could also be more convincing; there are many assays that could be used to show intracellular accumulation. While the evidence shown in this study supports AmvA polyamine transport, the use of whole cell accumulation assays does not provide direct evidence that AmvA transports the polyamines; it is an indirect association. Direct biochemical assays and structural studies would be required to conclusively show this. Finally, the discussion should include a reference to how these different pumps (aceI, AdeABC, AmvA) are collectively functioning as a system; the diagram in Fig. 3 could be moved to the discussion and a schematic devised to describe the proposed mechanism.

Major:

Figure 1B is not very informative. Would be useful to include GOC on the heat map.

Figure 2: Why does spermine repress many more genes than the other PAs? Figure legend and/or methods section should specify if these were biological replicates for B and C.

Line 141. This information should be included in the main text to support the plasma survival and biofilm assays. It's not immediately clear why serum resistance was assessed, this could be better explained/integrated.

Figure 3: (A) aceI shows an increase in expression in the presence of spermine; is this consistent with the previous study that showed enhanced expression with only putrescine and cadaverine? The shading is a little misleading because it looks like spermine provides a fold change of >4; the text says 2-4? Spermidine previously also showed an aceI 5-fold induction? In addition, line 185 claims that longer-chain polyamines are not aceI substrates, but the data shows induction with spermine, which is longer than the other polyamines? This should be clarified. Perhaps a bar chart would be more appropriate? The diagram on the right requires explanation.

(B) Growth should be shown for 0 mg/mL on the MIC chart. In addition, did the authors assess whether deletion of these genes impacted growth? A growth curve should be included. L200, why did deletion of amvR only increase the MIC (marginally) for spermine; the fold expression for both polyamines appear to be equal? The MIC data should be shown for putrescine and cadaverine too, especially considering all polyamines tested appear to enhance expression of amvR.

Line 190 – why would both the amvA pump and the repressor be induced?

Line 207 – substantiate this with either a reference or an explanation of how the analysis was performed and which genomes were assessed and why.

Line 226 – considering the intracellular levels of polyamines are very high under normal circumstances, how does this experiment compare in terms of intracellular accumulation? Is it physiologically relevant? The authors acknowledge the accumulation was very low, and this experiment was performed with a significant proportion of cells. Mass spec-based accumulation assays could be used.

Line 227 – why was the spermine MIC not shifted in BL21?

Line 237 – Please clarify the meaning of 'The strain background (WT or tn-amvR) was identified as significant source of variation by two-way ANOVA, $p < 0.0001$ '. Was the p -value < 0.0001 when amvR was deleted?

A side note: are there polar effects due to deletion of these genes? Have they been complemented?

Figure 4: This section (L227-247)/the reporter approach and the figure need attention. The basis of the reporter is not clear. Confusing use of acronyms (e.g., SPD etc). S-phase? *** is not over a data point. The panel E is completely missing/miss-labelled?

L251 – show data for this in the Supplementary section. The thermal shift data is not very convincing; polyamines are charged molecules that bind to many macromolecules. Could other approaches be used? Incubation and the denaturation to show release? ITC, SPR etc. Overall, this section could be significantly improved.

Minor:

L12 – fully explain AMR for the first use.

L33, L39-42, 51, 67, 177– references required

L48 – which pumps confer disinfectant resistance in particular?

L87 – define shock? This was sub-inhibitory.

L92 – why was 1/8 MIC selected?

L179 – define the context of 'known'

L217 – it would be useful to provide information on these strains to assist readers not overly familiar with *A. baumannii* strains. Would other strains be impacted by loss of amvA? Or would they have pumps that could compensate for polyamine efflux?

Line 228 – define MDE

Reviewer #2 (Remarks to the Author):

The study describes the identification of natural substrates for a drug efflux pump in the clinically important pathogen *Acinetobacter baumannii*. The pump has previously been implicated in mediating resistance to biocidal compounds, but the specificity of the pump for naturally occurring compounds has not been elucidated. The paper describes long chain polyamines as substrates for the pump and details the specificity of the pump and its regulator protein, which binds to the polyamines to de-repress expression. The work seems to be well designed and sufficient information is available for other researchers to be able to repeat.

The paper addresses a general question related to the natural substrates that would have been transported through such efflux pumps, before antibiotics and antiseptics were in common use. This is, in general, poorly understood and the paper provides important new information which relates directly to the role of one such pump in *Acinetobacter* survival and potentially pathogenesis. As such, I think the paper merits publication but could be strengthened by addressing some minor questions. On page 9, the authors identify that a transposon mutant in amvR results in a 2-fold increase in MIC. Understanding the increase in expression levels for the co-regulated amvA gene in this strain would be useful.

The authors note that that "the amvA repressor gene amvR25 was also induced by polyamine shock, however the cognate regulators of aceI and adeABC (aceR and adeRS, respectively) were not" Is this surprising given that AdeRS is a well-known regulator of AdeABC ?. Does the strain have a mutation in AdeS that means that adeABC is already overexpressed (as in the related strain AYE). Similarly, for

the cognate regulator of AceI

Could the authors please clarify the results shown in figure 3B. The panel with spermine shows a clear increase in resistance for the amvR transposon, but both the amvA and adeb transposons increase susceptibility to a similar degree, despite the presence of an intact copy of the other efflux pump. This would seem to be counterintuitive and needs explanation. In contrast, with spermidine, the amvR transposon has no effect on growth compared to the wild type. But the amvA transposon gives a significant increase in susceptibility, with the adeb-transposon having a lesser effect. Again could the authors please offer some explanation.

The authors identify that mutations in smvR which regulates the equivalent pump in Klebsiella and Proteus is linked to elevated chlorhexidine resistance and it would be good to see MIC data for chlorhexidine added to table 1. Understanding whether the mutations in these genes affect susceptibility to any different classes of antibiotics would also be useful. Although it might not be expected that antibiotics would be substrates for amrA, there is evidence for a role for polyamines in susceptibility to antibiotics (e.g. aminoglycosides in Pseudomonas PMID: 31383668) and this might be indirectly influenced by efflux of such through this pump.

In the discussion, could the authors please include some information about whether polyamine efflux pumps are known in other species and whether AmvA homologs in other species (as cited) might be expected to perform a similar function based on homology with AmvA.

My only real criticism is that the work focusses on a single strain of Acinetobacter from International clonal lineage 1. Although the operon is highly conserved and the choice of strain is dictated by the availability of transposon mutants in this background, it would be good to see at least some of the experimentation repeated in other strain backgrounds, notably in an a clonal lineage 2 background. Differences might reflect other aspects of polyamine metabolism which would be good to confirm independently.

Reviewer #3 (Remarks to the Author):

The study by Short et al examines the transcriptional responses to polyamine compounds in the nosocomial pathogen *A. baumannii* and identify efflux systems as important components of the response. Genetic and biochemical data are provided that test the model that the AmvR-AmvA system responds to and exports long-chain polyamines. The paper is well-written and addresses an important problem in understanding the function of efflux pumps in *Acinetobacter*. To address some weaknesses in the manuscript I have these suggestions for improvement:

Major points:

1) Currently the genetic data showing that spermidine and spermine act through AmvR to induce AmvA are not complete. The use of an amvR knockout and the amvA-GFP reporter in Fig. 4 is a good approach, but there is switching between strain backgrounds in different steps of the experiment. A rationale for switching is provided, but nonetheless this leaves the possibility that the reporter is spermidine-responsive in one background but not in the other. To make this analysis complete, a single strain background should be used to determine the ability of spermidine/spermine to enhance amvA expression, and the dependence on amvR for this response. If this test is not possible due to incompatibility with the reporter plasmid, another approach, such as qPCR, could be used to perform the complete analysis in one strain background.

2) Transposon mutant phenotypes (Fig. 3B,D, 4A) should be tested for complementation by the WT genes to rule out potential polar effects and second-site mutations that may be complicating the

phenotypes.

3) line 21-22, "these molecules induce expression of amvA through binding to its cognate regulator AmvR" seems like an overstatement. The data show that long-chain polyamines induce amvA expression, and support that they may be AmvR ligands, but connecting polyamine-AmvR binding to altered regulatory activity of AmvR was not shown. This would require demonstration that AmvR bound to long-chain polyamines shows altered activity (e.g., affinity for amvA promoter DNA). The statement should be rewritten in the absence of this data.

4) The differential scanning fluorimetry experiment should be clarified. Was this effect dose-dependent? What molarity does 0.2% polyamine salts correspond to?

5) Since amvA confers a greater degree of resistance to spermine/spermidine in *A. baumannii* vs *E. coli* (Fig. 3B vs Fig. 3E), a model that should be considered is that amvA collaborates synergistically with another *A. baumannii* pump, with the observed high-level resistance requiring both. AdeABC is the obvious candidate, since adeB contributes to spermine/spermidine resistance in *A. baumannii*. Can overexpressed amvA cause increased spermine or spermidine resistance in *A. baumannii*, and does this depend on adeABC?

Minor points:

-Can addition of sub-MIC spermidine or spermine enhance resistance to other AmvR or AdeABC substrates?

-In line 180/Fig. 3, adeABC are noted to show increased expression with polyamines. Leus et al (2020) showed that AB5075 has a SNP in the adeS control gene associated with increased adeABC expression. Does adeABC show increased expression with polyamines in a strain having a WT adeRS not associated with constitutive adeABC over expression?

-The timing of measurements in Fig. 4A is unclear. The labels include "overnight" and "S-Phase", but the Methods section says cultures were grown overnight or for 6h. "S-Phase" should be clearly defined, and the time points used for the expression data should be clarified.

-In Fig. 4, the acronyms (AD, PUT or SPD) should be defined in the figure legend.

-Transposon mutants should be described in more detail in the methods section/Table S1. In the table the mutants are listed as tn-amvA, tn-amvR, and tn-adeB, but specifics on the exact mutant/transposon position are not provided.

-In Table S3, what the color highlighting signifies should be clearly stated.

Referee expertise:

Referee #1: antimicrobials and bacterial efflux pumps

Referee #2: efflux pumps and antibiotic resistance

Referee #3: *A. baumannii* physiology/genetics

Reviewers' comments:

Reviewer #1 (Remarks to the Author):

Short et al., describe a thorough study intending to delineate the cellular transcriptional response of *A. baumannii* to high levels of polyamines. Through this process, they newly insinuate AmvA, and adeABC, as spermidine/spermine efflux pumps (long-chain polyamines). The impact of such a finding is significant since there are only a handful of drug efflux pumps that have been associated with this phenomenon. Furthermore, the link between polyamines and disinfectant resistance is compelling in terms of evolutionary significance. Below I have provided feedback to improve the study (L=line). Overall, the transcriptomic approach is a strength, and the evidence supporting AmvA's association with polyamine export is compelling. I have concerns regarding the suggestion that polyamines bind to AmvR, and thus alleviate the protein's repression of AmvA. While this may indeed be the case, this section of the study needs attention (as described below) and likely additional investigation

before this is conclusively stated. The assay involving spermidine accumulation could also be more convincing; there are many assays that could be used to show intracellular accumulation. While the evidence shown in this study supports AmvA polyamine transport, the use of whole cell accumulation assays does not provide direct evidence that AmvA transports the polyamines; it is an indirect association. Direct biochemical assays and structural studies would be required to conclusively show this. Finally, the discussion should include a reference to how these different pumps (aceI, AdeABC, AmvA) are collectively functioning as a system; the diagram in Fig. 3 could be moved to the discussion and a schematic devised to describe the proposed mechanism.

Major:

1. Figure 1B is not very informative. Would be useful to include GOC on the heat map.

We have moved Figure 1B to a supplementary figure. Since the expression changes are clustered to show the similarities and differences between the regulatory effects of each polyamine there is no straightforward way to add COG designations to this figure without losing other information (we assume this is what the reviewer means), however this information is easy for readers to find in the supplementary tables.

2. Figure 2: Why does spermine repress many more genes than the other PAs?

We do not wish to speculate on this in the manuscript as there is very little known about how polyamines are integrated into *A. baumannii* physiology. We hypothesise that the differences in the number of regulated genes could relate to different half-lives of these molecules in *A. baumannii* (due to differences in degradation pathways for each molecule), or that high spermine levels may stress particular metabolic pathways that then have downstream effects on expression of many other genes.

3. Figure legend and/or methods section should specify if these were biological replicates for B and C.

We have added this information to the figure legend, and thank the reviewer for pointing out this omission.

4. Line 141. This information should be included in the main text to support the plasma survival and biofilm assays. It's not immediately clear why serum resistance was assessed, this could be better explained/integrated.

Serum resistance and biofilm formation were assessed simply to give a more complete picture of the effects of polyamines on *A. baumannii* and the extent to which different polyamines elicit distinct

effects. However, we agree with the reviewer that these results are not well integrated, and we have moved them to the supplementary information and slightly changed the description of the results (lines 170-172).

5. Figure 3: (A) *aceI* shows an increase in expression in the presence of spermine; is this consistent with the previous study that showed enhanced expression with only putrescine and cadaverine?

We have improved the writing in this section to more accurately describe the RNAseq results and compare with the findings reported in Hassan *et al* 2020. Specifically:

- Hassan *et al* showed that all four polyamines induce *aceI* expression (spermine and spermidine weakly), but only putrescine and cadaverine are substrates.
- Our RNAseq results showed induction of *aceI* by putrescine and cadaverine (both log₂FC ~8) and spermine (log₂FC 4.85), while spermidine showed a log₂FC of 2.1 which failed the significance threshold in the RNAseq analysis.

The relevant section of the manuscript now reads (lines 192-196):

“The PACE family transporter gene *aceI* was the most dramatically upregulated, with a log₂-fold change of ~8 in the presence of putrescine and cadaverine and ~5 with spermine, while spermidine showed a weak (log₂-fold change of 2) induction that was not statistically significant. This fits with the recent finding that short chain diamines (including putrescine and cadaverine) are physiological substrates of AceI, while the longer-chain polyamines spermine and spermidine are not substrates but do weakly induce *aceI* expression (11).”

6. The shading is a little misleading because it looks like spermine provides a fold change of >4; the text says 2-4? Spermidine previously also showed an *aceI* 5-fold induction? In addition, line 185 claims that longer-chain polyamines are not *aceI* substrates, but the data shows induction with spermine, which is longer than the other polyamines? This should be clarified.

Perhaps a bar chart would be more appropriate? The diagram on the right requires explanation.

Our rewritten results section should clear up any ambiguity as we now differentiate between previous results regarding expression and those specifically considering transport. We do not believe our results contradict those reported in Hassan *et al* – the two studies used different approaches to determine *aceI* gene expression changes – qRT-PCR by Hassan *et al* and RNAseq here. Both studies find strong induction of *aceI* by putrescine and cadaverine and weak induction by spermine. The effect of spermidine on *aceI* expression cannot be reliably assessed from our data due to this result failing the statistical threshold.

We have changed the shading of the heatmap (now 2B) to a viridis colour scale to communicate the expression fold-changes more accurately. In our opinion using a bar chart may confuse readers as heatmaps are more typically used for RNAseq data. We have removed the diagram on the right.

(B) Growth should be shown for 0 mg/mL on the MIC chart. In addition, did the authors assess whether deletion of these genes impacted growth? A growth curve should be included. L200, why did deletion of *amvR* only increase the MIC (marginally) for spermine; the fold expression for both polyamines appear to be equal? The MIC data should be shown for putrescine and cadaverine too, especially considering all polyamines tested appear to enhance expression of *amvR*.

We have added no-substrate control growth readings to all MIC charts, and added results with putrescine and cadaverine to a supplementary figure (Figure S2A). We speculate that the *amvR* mutant showed an MIC increase with spermine but not spermidine because the spermidine MIC in AB5075 is so extremely high (40 mg/ml) that additional increases may not be possible due to biophysical constraints on the amount of substrate that can be removed by AmvA. We also speculate that at this high concentration of spermidine *amvA* expression is likely to be fully derepressed, such that deletion of *amvR* has no effect on *amvA* expression. In contrast spermine shows a more graded effect on growth with reduction of WT growth at 5mg/ml and 10mg/ml and complete inhibition at 20mg/ml. It is possible that the increased expression of *amvA* in an *amvR* mutant results in an equivalent increase in spermine and spermidine efflux capacity, but that this increase only leads to a change in MIC in the context of the lower spermine concentrations that inhibit *A. baumannii* growth. We have added the following sentence to the manuscript to clarify this point (lines 241-249):

“Mutation of *amvR*, which was previously shown to increase *amvA* expression by ~6-fold³⁴, increased the MIC of spermine 2-fold from 10 mg.ml⁻¹ to 20 mg.ml⁻¹ but had no effect on growth in the presence of spermidine. Note that the *A. baumannii* AB5075 spermidine MIC was extremely high (40 mg.ml⁻¹) – it is possible that the *amvR* mutant does have increased AmvA-mediated spermidine efflux, but that this is not sufficient to increase the MIC above 40 mg.ml⁻¹. Alternatively, *amvA* expression may be fully derepressed with 40 mg.ml⁻¹ spermidine such that mutation of *amvR* does not further increase expression”

7. Line 190 – why would both the *amvA* pump and the repressor be induced?

The *amvR* repressor and *amvA* are adjacent genes which are divergently transcribed; it is likely that AmvR repression affects both genes though we have not investigated this directly. In contrast, AceR and AdeRS are both transcriptional activators, so the model of common substrate-dependent derepression does not apply to these proteins. We have modified the text as follows (Lines 205-208):

“The *amvA* repressor gene *amvR* (25) is encoded 125-bp upstream of *amvA* on the opposite strand and was also induced by polyamine shock, however the cognate regulators of *aceI* and *adeABC* (*aceR* and *adeRS*, respectively, both transcriptional activators) were not.”

8. Line 207 – substantiate this with either a reference or an explanation of how the analysis was performed and which genomes were assessed and why.

We have expanded on the description of the pangenome analysis in the text, and provided the full gene presence-absence table (which includes NCBI identifiers for the assemblies used) as supplementary Table 5. The pangenome search was restricted to complete *A. baumannii* genomes in order to minimise the effects of contig boundaries on gene neighbourhood analysis. All complete genomes were used to give a reasonable representation of the diversity of the species. Although this genome set is biased towards clinical isolates and ICL-2 strains, it also includes non-clinical isolates.

9. Line 226 – considering the intracellular levels of polyamines are very high under normal circumstances, how does this experiment compare in terms of intracellular accumulation? Is it physiologically relevant? The authors acknowledge the accumulation was very low, and this experiment was performed with a significant proportion of cells. Mass spec-based accumulation assays could be used.

There is very little information on expected intracellular levels of spermidine in *A. baumannii* – the existing literature on endogenous polyamines in this species only identifies spermidine some of the time, suggesting that it is variable depending on environment and/or strain. Uptake of spermidine from the environment is expected to be low due to the hydrophobicity of this molecule. However, given the unparalleled sensitivity of transport assays using radiolabelled compounds and liquid scintillation, we believe that this was the best approach to measure transport. As this reviewer stated, intracellular accumulation assays only provide indirect evidence for transport. We attempted direct transport assays using purified AmvA in liposomes, however these were unsuccessful due to precipitation of AmvA during reconstitution. Since MFS transport proteins are very well known to be self-sufficient transport proteins, we believe that it is very likely that AmvA is directly responsible for the transport of spermidine. Furthermore, we have extensive evidence from a range of assays, including very strong phenotypic evidence (≥ 32 -fold MIC decrease in two different *A. baumannii* strains, reverse phenotype on *E. coli* overexpression) coupled with our evidence for specific regulation through AmvR.

10. Line 227 – why was the spermine MIC not shifted in BL21?

E. coli already has high tolerance to polyamines, which can mask effects of individual efflux pumps in this type of experiment. For example, the study identifying spermidine as a substrate of MdtIJ in *E. coli* used a spermidine acetyltransferase mutant for tolerance experiments, presumably because an effect was not seen in wild-type *E. coli*. We speculate that AmvA makes only a marginal contribution to clearance of spermine (through either efflux or catabolism) in *E. coli* because the endogenous clearance systems for this molecule are still operational.

11. Line 237 – Please clarify the meaning of ‘The strain background (WT or tn-amvR) was identified as significant source of variation by two-way ANOVA, $p < 0.0001$ ’. Was the p -value < 0.0001 when *amvR* was deleted?

In the revised version of the manuscript this data has largely been replaced by qRT-PCR data, see also our response to query on Figure 4.

12. A side note: are there polar effects due to deletion of these genes? Have they been complemented?

We did attempt to complement the *amvA* and *amvR* mutations. Unfortunately our complementation efforts were unsuccessful, because the vectors were apparently toxic to *A. baumannii* and did not yield any transformants despite multiple attempts (while other transformations conducted in parallel were successful). Regarding polar effects, neither *amvA* nor *amvR* appear to be in an operon with other genes. Instead of complementation we have provided additional lines of evidence for the function of these genes in polyamine export and polyamine-dependent regulation, including validation in an independent strain background and by overexpression in *E. coli* (*AmvA*), and ligand binding studies using purified protein (*AmvR*). Plasmid-based reporter expression assays in the *amvR* mutant background were conducted in addition to qPCRs (Hassan *et al* 2016, and new data in this revision) in order to exclude the possibility that the increased expression of *amvA* is caused by transcriptional readout from the transposon insertion in the *amvR* gene.

13. Figure 4: This section (L227-247)/the reporter approach and the figure need attention. The basis of the reporter is not clear. Confusing use of acronyms (e.g., SPD etc). S-phase? *** is not over a data point. The panel E is completely missing/miss-labelled?

We have added qPCR data to quantify *amvA* expression in an *amvR* mutant, and following induction with polyamines (Fig 4A), and this is now the primary experiment exploring *amvA* expression. Although our gene expression reporter experiments were not optimal due to vector toxicity and high background, we have kept these findings in the manuscript as Supplementary Figure 4 as we feel that having an independent line of evidence regarding *AmvR* and polyamine-dependent *amvA* regulation is still valuable. We have improved the presentation of the reporter assay results by 1) reporting only the *AmvR* vs WT reporter activity measured at OD = 2 to avoid confusion. 2) Using full polyamine names instead of acronyms, and 3) Describing the reporter construct architecture in the text (eg. lines 277-279). The *** is not over a data point because the mixed-repeated measures ANOVA tests for statistically significant differences between whole data series and not readings at a single time point; this is now elaborated on in the figure legend.

14. L251 – show data for this in the Supplementary section. The thermal shift data is not very convincing; polyamines are charged molecules that bind to many macromolecules. Could other approaches be used? Incubation and the denaturation to show release? ITC, SPR etc. Overall, this section could be significantly improved.

We have added the SEC data as Figure S4A, and have performed additional experiments to determine the binding affinity of *AmvR* for spermine, spermidine and putrescine by nanoDSF. The full nanoDSF data for determining polyamine binding affinity is reported in Figure 4B, Figure S4C and Supplementary tables 6 and 7. The low μ M affinities measured for spermine and spermidine suggest a specific interaction.

Minor:

1. L12 – fully explain AMR for the first use.

We have changed this to “Antimicrobial resistance”

2. L33, L39-42, 51, 67, 177– references required

Most of the statements in these sections were already referenced, but we have changed the position and number of some references to make it clearer which reference applies to which statement in multi-clause sentences, and added several references on efflux-mediated antibiotic resistance in *A. baumannii*.

3. L48 – which pumps confer disinfectant resistance in particular?

Specific details have been added (lines 56-58), along with appropriate references. Note this is not an exhaustive list and some MDEs have not been tested for disinfectant resistance activity.

4. L87 – define shock? This was sub-inhibitory.

We believe the use of “shock” is appropriate in this case because the experiment involved RNA sequencing soon after exposure to high levels of exogenous polyamines (at the upper end or above physiological concentrations), without substantial time for any adaptive responses to have an effect post-transcriptional changes. The full details of the experiment are provided for readers to avoid confusion.

5. L92 – why was 1/8 MIC selected?

1/8 MIC was selected as we hypothesised that growth/survival at this concentration would require induction of polyamine efflux systems without causing excessive stress on the cell.

6. L179 – define the context of ‘known’

We have removed this word as it was confusing and each of these efflux pumps is now mentioned in the introduction.

7. L217 – it would be useful to provide information on these strains to assist readers not overly familiar with *A. baumannii* strains. Would other strains be impacted by loss of *amvA*? Or would they have pumps that could compensate for polyamine efflux?

We have removed the reference to strain AC0037 as this sentence may not have communicated our intended meaning. Previous reports of the *AmvA* substrate range used *E. coli* overexpression tests and an *amvA* mutant of a more drug-sensitive strain, and we wished to state that the results of our MIC tests in AB5075 findings do not contradict these. The second point is now addressed by additional experiments with an *amvA* mutant of *A. baumannii* BAL062 (a strain from international clonal complex 2). We have added information at various points in the manuscript (lines 52, 264, 345) to state that there are two major circulating clinical lineages of *A. baumannii*, and we have tested *AmvA* activity in representatives of both lineages.

8. Line 228 – define MDE

This is now defined in the introduction

Reviewer #2 (Remarks to the Author):

The study describes the identification of natural substrates for a drug efflux pump in the clinically important pathogen *Acinetobacter baumannii*. The pump has previously been implicated in mediating resistance to biocidal compounds, but the specificity of the pump for naturally occurring compounds has not been elucidated. The paper describes long chain polyamines as substrates for the pump and details the specificity of the pump and its regulator protein, which binds to the polyamines to de-repress expression. The work seems to be well designed and sufficient information is available for other researchers to be able to repeat.

The paper addresses a general question related to the natural substrates that would have been transported through such efflux pumps, before antibiotics and antiseptics were in common use. This is, in general, poorly understood and the paper provides important new information which relates directly to the role of one such pump in *Acinetobacter* survival and potentially pathogenesis. As such, I think the paper merits publication but could be strengthened by addressing some minor questions.

1. On page 9, the authors identify that a transposon mutant in *amvR* results in a 2-fold increase in MIC. Understanding the increase in expression levels for the co-regulated *amvA* gene in this strain would be useful.

We have added the specific fold change information from previous work to line 217, and this point is also addressed by the inclusion of new qPCR data as suggested. However, in the MIC experiment *amvA* will also be induced by its polyamine substrates so the marginal impact of *amvR* loss may be lower.

2. The authors note that that “the *amvA* repressor gene *amvR25* was also induced by polyamine shock, however the cognate regulators of *aceI* and *adeABC* (*aceR* and *adeRS*, respectively) were not” Is this surprising given that *AdeRS* is a well-known regulator of *AdeABC* ?. Does the strain have a mutation in *AdeS* that means that *adeABC* is already overexpressed (as in the related strain AYE). Similarly, for the cognate regulator of *AceI*

Our current model for *AmvR* function is that it is a ligand-binding repressor (where binding of the ligand triggers de-repression). *AdeRS* and *AceR* are both transcriptional activators and are not necessarily expected to regulate their own expression in addition to that of their targets. While we feel that a full description of these transcription factors is secondary to the message of this study, we have added additional information on these at lines 205-208. We have also added a sentence about constitutive expression of *adeABC*. This section now reads:

“The *adeABC* genes, encoding an RND family transporter, showed a 2-3 log₂-fold increase in expression in the presence of any of the four molecules. This apparent induction by polyamines was surprising given that *A. baumannii* strain AB5075 carries a mutation in the regulator *AdeS* which causes constitutive *adeABC* expression^{32,33}. The MFS transporter gene *amvA* was induced by spermidine and spermine but not by putrescine and cadaverine. The *amvA* repressor gene *amvR*³⁴, which is encoded 125-bp upstream of *amvA* on the opposite strand, was also induced by polyamine shock, however the cognate regulators of *aceI* and *adeABC* (*aceR* and *adeRS*, respectively, both transcriptional activators) were not.”

3. Could the authors please clarify the results shown in figure 3B. The panel with spermine shows a clear increase in resistance for the *amvR* transposon, but both the *amvA* and *adeB* transposons increase susceptibility to a similar degree, despite the presence of an intact copy of the other efflux pump. This would seem to be counterintuitive and needs explanation. In contrast, with spermidine, the *amvR* transposon has no effect on growth compared to the wild type. But the *amvA* transposon gives a significant increase in susceptibility, with the *adeB*-transposon having a lesser effect. Again could the authors please offer some explanation.

The contribution of any efflux pump to resistance or tolerance to its substrates will depend on its expression level, transport rate, affinity for its substrates and the presence of other efflux pumps with overlapping specificities. We do not believe our MIC findings are counterintuitive – *AmvA* and *AdeABC* appear to have overlapping substrate specificities, and neither pump has a sufficient export capacity to remove very high amounts of polyamines on its own. This effect is clearer with spermidine than with spermine, where the presence of *AdeABC* without *AmvA* results in close to wild-type levels of resistance. We speculate that the reason the *amvR* mutant shows increased resistance to spermine but not spermidine is a consequence of the higher spermidine MIC (40mg/ml compared to 10mg/ml), meaning that an extremely high difference in efflux rate would be needed to result in an MIC increase. It is also likely that *amvA* expression is fully derepressed in the presence of 40mg/ml spermidine, such that loss of *amvR* does not result in further increases in expression. We have elaborated on this point in lines 219-224. While some cooperation between these systems is a possibility, we believe it is outside the scope of the present study which focuses on *AmvA*.

The authors identify that mutations in *smvR* which regulates the equivalent pump in *Klebsiella* and *Proteus* is linked to elevated chlorhexidine resistance and it would be good to see MIC data for chlorhexidine added to table 1. Understanding whether the mutations in these genes affect susceptibility to any different classes of antibiotics would also be useful. Although it might not be expected that antibiotics would be substrates for *amrA*, there is evidence for a role for polyamines in susceptibility to antibiotics (e.g. aminoglycosides in *Pseudomonas* PMID: 31383668) and this might be indirectly influenced by efflux of such through this pump.

We have added chlorhexidine MIC data to Table 1 and thank the reviewer for the useful suggestion. *AmvA* was previously shown to contribute to resistance to several antibiotics (Rajamohan *et al* 2010, *JAC*), however we could not reproduce these findings in preliminary tests, presumably because AB5075 has very high intrinsic drug resistance, and did not continue and test these with experimental

replicates. We consider it likely that exogenous polyamines may influence antibiotic susceptibility in *A. baumannii* AB5075 due to their substantial impact on expression of metabolic genes, and their induction of efflux pumps. We believe that this question merits detailed investigation in a separate study.

In the discussion, could the authors please include some information about whether polyamine efflux pumps are known in other species and whether AmvA homologs in other species (as cited) might be expected to perform a similar function based on homology with AmvA.

Only a handful of bacterial transporters of polyamines are known and we have referred to this work in the introduction and discussion. Blt of *B. subtilis* is (to our knowledge) the only other MFS family protein known to transport polyamines in bacteria. We speculate that the SmvA proteins cited have polyamine transport activity, while QacA is more distantly related, however our attempts to test these in an overexpression system were unsuccessful due to toxicity of the constructs. We have added information on sequence identity between SmvA, QacA and AmvA as we agree that readers may find this interesting, the section now reads:

“Interestingly, multiple bacterial pathogens including *Staphylococcus aureus*, *Proteus mirabilis* and members of the *Enterobacteriaceae* possess AmvA homologues (QacA, SmvA) which, like AmvA, confer increased biocide resistance (particularly to chlorhexidine) and are upregulated or have increased prevalence in clinical strains (40–43). AmvA shares 50-55% amino acid identity with the enterobacterial SmvA proteins, while *S. aureus* QacA is more distantly related (30% amino acid identity). It is tempting to speculate that some of these clinically-important AmvA homologues may also have physiological roles in polyamine transport.”

My only real criticism is that the work focusses on a single strain of *Acinetobacter* from International clonal lineage 1. Although the operon is highly conserved and the choice of strain is dictated by the availability of transposon mutants in this background, it would be good to see at least some of the experimentation repeated in other strain backgrounds, notably in an a clonal lineage 2 background. Differences might reflect other aspects of polyamine metabolism which would be good to confirm independently.

This is a valuable point, and we have constructed and tested an *amvA* deletion mutant in an ICL2 strain, BAL062. This mutant showed a similar reduction in spermidine and spermine resistance to the equivalent mutant in AB5075.

Reviewer #3 (Remarks to the Author):

The study by Short et al examines the transcriptional responses to polyamine compounds in the nosocomial pathogen *A. baumannii* and identify efflux systems as important components of the response. Genetic and biochemical data are provided that test the model that the AmvR-AmvA system responds to and exports long-chain polyamines. The paper is well-written and addresses an important problem in understanding the function of efflux pumps in *Acinetobacter*. To address some weaknesses in the manuscript I have these suggestions for improvement:

Major points:

1) Currently the genetic data showing that spermidine and spermine act through AmvR to induce AmvA are not complete. The use of an *amvR* knockout and the *amvA*-GFP reporter in Fig. 4 is a good approach, but there is switching between strain backgrounds in different steps of the experiment. A rationale for switching is provided, but nonetheless this leaves the possibility that the reporter is spermidine-responsive in one background but not in the other. To make this analysis complete, a single strain background should be used to determine the ability of spermidine/spermine to enhance *amvA* expression, and the dependence on *amvR* for this response. If this test is not possible due to incompatibility with the reporter plasmid, another approach, such as qPCR, could be used to perform the complete analysis in one strain background.

We have performed qPCR experiments in AB5075 as suggested, to examine induction of *amvA* expression by long-chain polyamines, and the dependence of this induction on AmvR. This is now

reported in Figure 4A. In our preliminary AB5075 reporter experiments we did find modest spermidine-dependent induction of the *amvA*_{prom}-GFP reporter but these were compromised by high background fluorescence, so we judged that a different reporter construct/strain background would be more reliable for assessing possible induction by putrescine and cadaverine (now Supplementary Figure 3C).

2) Transposon mutant phenotypes (Fig. 3B,D, 4A) should be tested for complementation by the WT genes to rule out potential polar effects and second-site mutations that may be complicating the phenotypes.

Our attempts to complement these mutants using plasmids were unsuccessful, as detailed in our response to Reviewer 1. However, we have provided additional evidence for the role of *amvA* in polyamine tolerance through overexpression in *E. coli*, and testing a deletion mutant in an independent strain background. The function of *amvR* has been explored through reporter assays (which are not likely to be affected by transposon-dependent polar effects on *amvA* transcription) as well as qPCR, and through biochemical experiments. For these reasons our conclusions are very unlikely to be compromised by polar-effects or secondary mutations.

3) line 21-22, "these molecules induce expression of *amvA* through binding to its cognate regulator AmvR" seems like an overstatement. The data show that long-chain polyamines induce *amvA* expression, and support that they may be AmvR ligands, but connecting polyamine-AmvR binding to altered regulatory activity of AmvR was not shown. This would require demonstration that AmvR bound to long-chain polyamines shows altered activity (e.g., affinity for *amvA* promoter DNA). The statement should be rewritten in the absence of this data.

We have rewritten this statement to more accurately reflect our data. We have also added nanoDSF experiments to determine the affinity of AmvR for polyamines. Because spermidine is also known to bind to DNA, we judged that an experiment examining the effect of long-chain polyamines on AmvR-*amvA* promoter binding would be extremely difficult to interpret.

4) The differential scanning fluorimetry experiment should be clarified. Was this effect dose-dependent? What molarity does 0.2% polyamine salts correspond to?

We agree that this was a weakness in the original manuscript, and have improved this section by performing nanoDSF to determine AmvR-polyamine binding affinity. The DSF experiments are still included in supplementary information, and we have reported the molar concentrations in each case.

5) Since *amvA* confers a greater degree of resistance to spermine/spermidine in *A. baumannii* vs *E. coli* (Fig. 3B vs Fig. 3E), a model that should be considered is that *amvA* collaborates synergistically with another *A. baumannii* pump, with the observed high-level resistance requiring both. AdeABC is the obvious candidate, since *adeB* contributes to spermine/spermidine resistance in *A. baumannii*. Can overexpressed *amvA* cause increased spermine or spermidine resistance in *A. baumannii*, and does this depend on AdeABC?

We do not wish to over-interpret the observation that overexpression of AmvA produced a (comparatively) modest spermidine resistance increase in *E. coli* because 1) heterologous overexpression of membrane proteins in *E. coli* is often slightly toxic, and 2) AmvA is operating on top of the endogenous polyamine catabolism and efflux systems of *E. coli*, which already has high intrinsic polyamine resistance. Our interpretation is simply that AmvA and AdeABC have overlapping substrate affinities and both are required for resistance to very high levels of polyamines. A cooperative model is likely given that tripartite and single component efflux pumps are known to provide cooperative efflux/resistance, and worth investigating, but we feel that to do this thoroughly (for example investigating resistance at multiple expression levels of each pump, examining cross-regulation between *adeABC* and *amvA*) would require a large body of experimental work that is more suited to a follow-up study.

Minor points:

-Can addition of sub-MIC spermidine or spermine enhance resistance to other AmvR or AdeABC substrates?

We have not tested this, but we speculate that this may be possible for AmvA.

-In line 180/Fig. 3, *adeABC* are noted to show increased expression with polyamines. Leus et al (2020) showed that AB5075 has a SNP in the *adeS* control gene associated with increased *adeABC* expression. Does *adeABC* show increased expression with polyamines in a strain having a WT *adeRS* not associated with constitutive *adeABC* over expression?

We have added a statement regarding higher expression of *adeABC* in AB5075. In general we feel that the role of *adeABC* in polyamine transport merits a separate study.

-The timing of measurements in Fig. 4A is unclear. The labels include "overnight" and "S-Phase", but the Methods section says cultures were grown overnight or for 6h. "S-Phase" should be clearly defined, and the time points used for the expression data should be clarified.

This figure has now been replaced by qPCR data and the reporter assay presentation has been improved. We now only report the 6h growth/OD = 2 measurement for WT vs *amvR* reporter expression.

-In Fig. 4, the acronyms (AD, PUT or SPD) should be defined in the figure legend.

This has been changed

-Transposon mutants should be described in more detail in the methods section/Table S1. In the table the mutants are listed as *tn-amvA*, *tn-amvR*, and *tn-adeB*, but specifics on the exact mutant/transposon position are not provided.

We have corrected this omission in Table S1.

-In Table S3, what the color highlighting signifies should be clearly stated.

We have removed the conditional formatting in Table S3 as it may not be retained if the file is opened using different software.

REVIEWERS' COMMENTS:

Reviewer #2 (Remarks to the Author):

I would like to thank the authors for their careful consideration of the points raised and the additional work that has gone into answering these. I am entirely satisfied with the responses and recommend publication of this important manuscript. I look forward to seeing future studies in this important area.

Reviewer #3 (Remarks to the Author):

The resubmitted manuscript is much improved and addresses most of my previous concerns.

Complementation experiments would have strengthened the argument that AmvA determines polyamine resistance in *A. baumannii*, but the analysis with an *amvA* knockout in a completely different isolate provides further support for their claim. The new data with AmvR provide additional support for their model that it functions as a polyamine-binding AmvA repressor.

I had the remaining minor points:

line 195 of the resubmission: "Fig S2B, S2C" should be S1B, S1C

lines 329-330 of the resubmission: "The *amvR* mutant strain showed a ~10-fold increase in expression of *amvA*" should specify that it is compared to WT control